# DNA passes through cohesin's hinge as well as its Smc3–kleisin interface

James E Collier, Kim A Nasmyth*

Department of Biochemistry, University of Oxford, Oxford, United Kingdom

**Abstract** The ring model proposes that sister chromatid cohesion is mediated by co-entrapment of sister DNAs inside a single tripartite cohesin ring. The model explains how Scc1 cleavage triggers anaphase but has hitherto only been rigorously tested using small circular mini-chromosomes in yeast, where covalently circularizing the ring by crosslinking its three interfaces induces catenation of individual and sister DNAs. If the model applies to real chromatids, then the ring must have a DNA entry gate essential for mitosis. Whether this is situated at the Smc3/Scc1 or Smc1/Smc3 hinge interface is an open question. We have previously demonstrated DNA entrapment by cohesin in vitro (Collier et al., 2020). Here we show that cohesin in fact possesses two DNA gates, one at the Smc3/Scc1 interface and a second at the Smc1/3 hinge. Unlike the Smc3/Scc1 interface, passage of DNAs through SMC hinges depends on both Scc2 and Scc3, a pair of regulatory subunits necessary for entrapment in vivo. This property together with the lethality caused by locking this interface but not that between Smc3 and Scc1 in vivo suggests that passage of DNAs through the hinge is essential for building sister chromatid cohesion. Passage of DNAs through the Smc3/Scc1 interface is necessary for cohesin's separase-independent release from chromosomes and may therefore largely serve as an exit gate.

## Editor's evaluation

This paper is a fundamental step forward in our understanding of how cohesin interacts with DNA to execute sister chromatid cohesion. Compelling biochemical evidence is presented to demonstrate the functional entry gate of cohesin for the topological entrapment of DNA. This important study will be of interest to researchers in the chromosome biology field, particularly those focused on SMC proteins and genome organisation.

*For correspondence:
ashley.nasmyth@bioch.ox.ac.uk

**Competing interest:** The authors declare that no competing interests exist.

## Introduction

The sister chromatid cohesion essential for mitosis and meiosis is mediated by a pair of rod-shaped SMC proteins (Smc1 and Smc3) joined together through an interaction between hinge domains at one end. The interconnection by a kleisin subunit (Scc1) of their ATPase domains at the other end creates a ring-like structure within which it is proposed sister DNAs are entrapped during DNA replication (*Haering et al., 2002*). Two approaches have hitherto been used to test the ring model. The first has been a method to induce thiol-specific chemical crosslinks within the three interfaces of SMC–kleisin (S–K) rings. By this means, it has been demonstrated that covalent circularization by bismaleimidoethane (BMOE) of a version of cohesin containing cysteine pairs within the ring's three interfaces is sufficient to cause catenation of small circular sister DNAs that are otherwise not inter-twined, hence proving their co-entrapment (*Haering et al., 2008*; *Gligoris et al., 2014*). The subsequent analysis of a wide variety of strains carrying different cohesin mutations confirmed that catenation in this manner of sister mini-chromosome DNAs correlates with the ability of yeast cells to proliferate (*Srinivasan et al., 2018*).

The second approach has been to elucidate the mechanism by which DNAs enter S–K rings. The logic being that only when we have understood this mechanism and found it to operate inside cells could we be certain that entrapment does indeed form the basis of cohesion. The initial goal was to establish which of the cohesin rings' three interfaces must open up to let in DNA, in other words to identify cohesin's DNA entry gate. The finding that interconnection of Smc1 and Smc3 hinges using rapamycin (when FKBP12 and FRB were inserted into small loops within the Smc1/3 hinges) blocked cohesion establishment while fusion of Smc3 or Smc1 with Scc1 did not do so led to the proposal that if cohesin has a unique essential DNA entry gate, then it must be at the hinge interface (*Gruber et al., 2006*). However, these experiments merely showed that a modification predicted to hinder hinge opening blocks establishment of cohesion, which is not the same as proving that DNAs actually enter via this interface. Besides which, the conclusion that the hinge is a DNA entry gate is not universally accepted (*Murayama and Uhlmann, 2015*; *Murayama et al., 2018*). Thus, despite being crucial for understanding how cohesion is established, the location of cohesin's DNA entry gate remains unresolved. To break this impasse, we recently developed an in vitro assay to measure entrapment of DNAs within S–K rings, using the same technique used in vivo, namely catenation of circular DNAs by cysteine-substituted cohesin rings chemically circularized using BMOE (*Collier et al., 2020*). S–K DNA entrapment in vitro is stimulated by Scc2 and depends on ATP and Scc3 but not on Pds5 or Wapl, reflecting the properties of S–K entrapment in vivo (*Srinivasan et al., 2018*), suggesting the reaction is physiologically relevant. Using this system, we now describe experiments that show definitively that cohesin possesses two gates through which DNAs pass, at least in vitro, one at the Smc3/Scc1 interface and a second at the Smc1/3 hinge. We also describe a series of topological assays suggesting that the first step is passage of DNAs between cohesin's ATPase heads and their enclosure by Scc2 in the lower half of an SMC chamber bounded by hinges and heads engaged by ATP (called the E-S compartment).

## Results

### Covalent closure of cohesin ring interfaces

Cohesin's DNA gate(s) could in principle be identified merely by observing the process of entrapment in real time. Because this is not at present technically possible, we instead covalently sealed each ring interface shut in a manner orthogonal to the thiol-specific crosslinking protocol that we use to

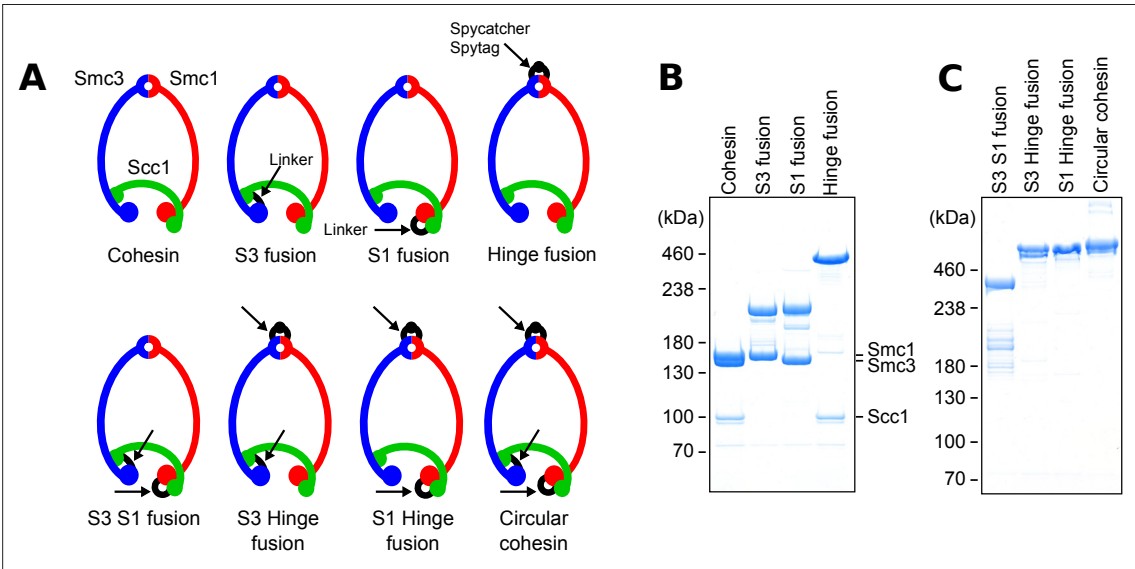

**Figure 1.** Covalent closure of cohesin's interfaces. (**A**) The cohesin ring and fusion variants. Covalently sealed interfaces are linked in black and are indicated with arrows. (**B**) Coomassie stain of purified cohesin with either the Smc3/Scc1 (S3 fusion), Scc1/Smc1 (S1 fusion), or hinge interfaces (Hinge fusion) covalently sealed. (**C**) Coomassie stain of purified cohesin with multiple interfaces covalently sealed.

The online version of this article includes the following figure supplement(s) for figure 1:

**Figure supplement 1.** Covalent closure of cohesin's interfaces.

measure entrapment (*Collier et al., 2020*). This approach allows us to seal potential DNA gates prior to the entrapment process. If DNA entered through a single gate, then sealing should block entrapment. Though such a result would be consistent with the gate being used for entrapment, it would not directly demonstrate passage, as the sealing process could in principle also interfere with passage through a different gate, by somehow altering the latter's conformation. Crucially, a failure to block entrapment by sealing a single ring interface would imply that the interface in question is either not a gate or that that it is not the sole entry gate. Though a striking result, blocking entry would therefore permit only weak conclusions. A more rigorous approach would be to seal two out of the ring's three interfaces, leaving only a single potential gate. In this case, continued entrapment would demonstrate that DNA must have passed through the sole remaining interface.

To seal the Smc3/Scc1 interface, we expressed a fusion protein (*Davidson et al., 2019*) in which the N-terminus of Scc1 is connected by a short linker to the C-terminus of Smc3 (S3 fusion; *Figure 1A*, *Figure 1—figure supplement 1A*). A similar approach was used connect Scc1's C-terminus to Smc1's N-terminus, thereby sealing the Smc1/Scc1 interface (S1 fusion). Smc1 and Smc3 hinges cannot be connected using this approach because they are located within the middle of their polypeptides. In this case, we achieved highly efficient covalent closure using an isopeptide bond created between a spytag and a spycatcher domain (*Li et al., 2014*) inserted into loops on the surface of Smc1 and Smc3 hinges, respectively (Hinge fusion; *Figure 1—figure supplement 1B*). To measure DNA entrapment within their S–K compartments after BMOE treatment, each of these S–K complexes contained a pair of cysteine pairs at their two unaltered interfaces. We next created three types of cohesin complexes composed of a single polypeptide in which two out of three interfaces were covalently sealed, namely a Smc3–Scc1–Smc1 (S3 S1 fusion) fusion containing a wild-type hinge interface, a Smc3–Scc1 fusion whose hinge was connected to that of Smc1 using a spytag–spycatcher pair (S3 Hinge fusion), and a Scc1–Smc1 fusion whose hinge was similarly connected to that of Smc3 (S1 Hinge fusion). In each case, the one remaining unaltered interface contained a pair of cysteines that could be crosslinked by BMOE. Lastly, we created a covalently circular ring in which all three interfaces were sealed (Circular cohesin). Remarkably, all seven types of cohesin rings produced abundant and stable complexes (*Figure 1B, C*). Some had modestly reduced rates of ATP hydrolysis, with circular cohesin having the greatest defect (~60% that of WT cohesin; *Figure 1—figure supplement 1C*).

## DNA passes through the Smc1/Smc3 hinge as well as the Smc3/Scc1 interface

We next tested the ability of these different fusion proteins to entrap circular plasmid DNA in vitro (*Figure 2A*; *Collier et al., 2020*). To measure DNA entrapment, we incubated cohesin with ATP, Scc2, Scc3, and circular DNA. Because all cohesin rings contain cysteine pairs within interfaces not sealed by a direct protein fusion, their S–K rings (*Figure 2B*) could subsequently be covalently circularized by addition of the crosslinking reagent BMOE. Following protein denaturation by heating at 70°C for 10 min in the presence of 1% sodium dodecyl sulfate (SDS), DNA protein mixtures were fractionated by agarose gel electrophoresis. Covalently circularized cohesin rings that have entrapped DNA retard the latter's electrophoretic mobility, causing a 'gel-shift'. Depending on its efficiency, the concatenation of DNA with cohesin gives rise to a ladder of retarded species, corresponding to successive entrapment by one, two, three, or more S–K rings.

As expected from in vivo results (*Gruber et al., 2006*; *Srinivasan et al., 2018*), DNAs were entrapped by the cohesin rings containing either the Smc3–Scc1 (S3) or Scc1–Smc1 (S1) fusions (*Figure 2C*). More surprising, they were also entrapped by rings with the Hinge fusion. It is difficult to make direct comparisons between the efficiencies as entrapment by each version depends on different cysteine pair combinations, whose crosslinking by BMOE can vary. We can nevertheless conclude that DNA enters the cohesin ring via at least two different gates. To identify these, we measured entrapment by rings with two interfaces sealed, whereby DNA can only enter through the one remaining open interface (*Figure 2D*). These assays revealed DNA entrapment by cohesin rings containing the Smc3–Scc1–Smc1 fusion (S3 S1 fusion) as well as rings containing the Scc1–Smc1–Hinge fusions (S1 Hinge fusion). However, DNA was not entrapped by a complex with the Smc3–Scc1–Hinge fusions (S3 Hinge fusion). These observations demonstrate that DNA entrapment arises by passage through the hinge as well as through the Smc3/Scc1 interface. Importantly, no passage occurs through the Smc1/Scc1 interface, at least when the other two gates are covalently sealed.

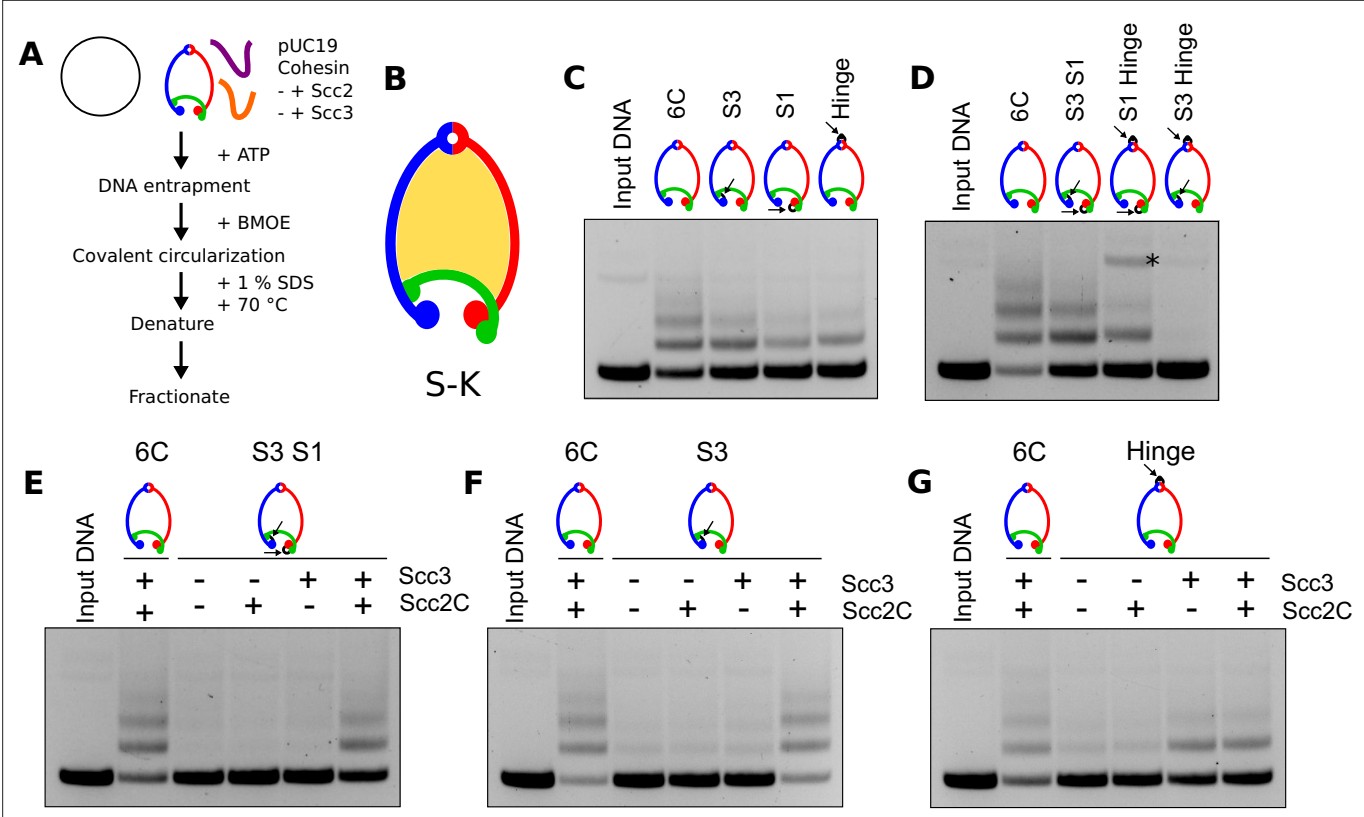

**Figure 2.** DNA passes through cohesin's hinge and Smc3/Scc1 interfaces. (**A**) Schematic of the in vitro DNA entrapment assay. (**B**) SMC–kleisin (S–K) ring. (**C**) DNA entrapment comparing WT cohesin (6C) with the Smc3–Scc1 (S3), Scc1–Smc1 (S1), or hinge (Hinge) fusion proteins in the presence of Scc2 and Scc3 after 40 min. (**D**) DNA entrapment comparing WT cohesin (6C) with the Smc3–Scc1–Smc1 (S3 S1), Scc1–Smc1–Hinge (S1 Hinge), or Smc3–Scc1–Hinge (S3 Hinge) fusions in the presence of Scc2 and Scc3 after 40 min. * = nicked DNA. (**E**) DNA entrapment comparing WT cohesin (6C) in the presence of Scc2 and Scc3 with the Smc3–Scc1–Smc1 (S3 S1) fusion in the presence or absence of Scc2 and Scc3 after 40 min. (**F**) DNA entrapment comparing WT cohesin (6C) in the presence of Scc2 and Scc3 with the Smc3–Scc1 (S3) fusion in the presence or absence of Scc2 and Scc3 after 40 min. (**G**) DNA entrapment comparing WT cohesin (6C) in the presence of Scc2 and Scc3 with the Hinge fusion construct in the presence or absence of Scc2 and Scc3 after 40 min.

The online version of this article includes the following figure supplement(s) for figure 2:

**Figure supplement 1.** DNA entrapment by SMC–kleisin (S–K) rings.

## Scc2 is required for DNA passage through the hinge but not through the Sm3/Scc1 interface

Entrapment within S–K rings in vivo normally depends on both Scc2 and Scc3. We therefore addressed whether hinge or Smc3/Scc1 gate passage in vitro shares this property. Entrapment by the Smc3–Scc1–Smc1 (S3 S1) fusion was abolished by omission of either regulatory protein (*Figure 2E*). Entrapment via the hinge in vitro therefore resembles in vivo entrapment (*Srinivasan et al., 2018*). This suggests that the reason why cohesion establishment is abolished by linkage of Smc1 and Smc3 hinges containing FRB and FKBP12, respectively, using rapamycin (*Gruber et al., 2006*) is because passage of DNA through the hinge is an essential step. Interestingly, the strict dependence of hinge-mediated entrapment on Scc2 differs from entrapment by wild-type S–K rings in vitro, which still occurs in the absence of Scc2, albeit at a reduced level (*Figure 2—figure supplement 1A*; *Collier et al., 2020*). A simple explanation for this difference is that cohesin entraps DNA via both hinge and Smc3/Scc1 pathways in vitro and the Scc2-independent entrapment is due entirely to passage through the Smc3/Scc1 gate. If so, sealing the Smc3/Scc1 gate should abolish Scc2-independent S–K entrapment. As predicted, entrapment by rings containing the Smc3–Scc1 (S3) fusion depends on Scc2 as well as Scc3 (*Figure 2F*). In other words, Scc2-independent entrapment requires a Smc3/Scc1 gate that can be opened.

To address whether entrapment via the Smc3/Scc1 gate is affected by Scc2, we measured the effect of Scc2 and Scc3 on entrapment of DNA by cohesin whose hinge alone is fused and contains cysteine pairs within both SMC–Scc1 interfaces. Because DNA cannot pass through the Smc1/Scc1 interface (*Figure 2D*), all entrapment by such cohesin must be via its Smc3/Scc1 interface. Notably, entrapment by this construct depends on Scc3 but not Scc2 (*Figure 2G*). This dependence of DNA passage through the Smc3/Scc1 interface on Scc3 but not Scc2 in vitro resembles the activity that dissociates cohesin from chromosomes in vivo, a process dependent on Wapl and also blocked by the Smc3–Scc1 fusion (*Chan et al., 2012*). It is therefore possible that our in vitro experiments capture this process, albeit acting in reverse, as previously suggested (*Murayama and Uhlmann, 2015*). In vivo, release not only does not require Scc2 but is actively blocked by it, at least in G1 cells (*Srinivasan et al., 2019*). Given that passage of DNA through the Smc3/Scc1 interface is not required for cell proliferation, for S–K entrapment in vivo, or even for cohesin's stable association with the bulk of the genome, it is uncertain whether passage of DNA through this gate has any role in building cohesion in addition to its well documented role in mediating release.

## DNAs entrapped in E-S compartments in the absence of Scc3 are located between Scc2 and engaged heads

Given that passage of DNA through the hinge may be essential for building cohesion, the strict dependence of this process on Scc2 raises a question as to Scc2's role. Reactions performed in the absence of Scc3 provide an important clue. Under these circumstances, Scc2 promotes rapid entrapment of DNA within cohesin's E-S compartment (*Figure 3D*), namely between the hinge and Smc1 and Smc3 head domains engaged in the presence of ATP (*Collier et al., 2020*). Crucially, this process is not accompanied by entrapment within S–K rings (*Collier et al., 2020*). Cryo-EM structures of DNA oligonucleotides bound to Scc2 and cohesin suggest that DNA trapped within E-S compartments by Scc2 binds simultaneously to Scc2 and a groove created by the engagement of Smc1 and Smc3 heads in the presence of ATP (PDB 6ZZ6). DNA associates with similar grooves above the engaged heads of condensin (*Lee et al., 2022*), MukBEF (*Bürmann et al., 2021*), and Rad50 (*Käshammer et al., 2019*), implying that this type of association is a highly conserved feature of SMC-like ATPase domains. In the case of cohesin, the DNA is actually 'clamped' in a small compartment created by association of Scc2's N-terminal and central domains bound to Smc3's neck and head domains, respectively (*Collier et al., 2020*; *Higashi et al., 2020*; *Shi et al., 2020*). Though the conditions under which these cryo-EM structures were obtained resemble those necessary for entrapment of DNA within the E-S compartment, namely both require Scc2, ATP, and DNA, but not Scc3 or ATP hydrolysis (*Collier et al., 2020*), we cannot be certain whether the two activities are truly synonymous. What is required is a crosslinking assay for DNA clamping comparable to the one used to measure E-S compartment entrapment.

We therefore designed a set of cysteine pairs within Scc2–SMC interfaces that could be crosslinked by BMOE if DNA were clamped in the manner observed in the cryo-EM structure (*Figure 3—figure supplement 1A*). To this end, cysteines were introduced into the interfaces between Scc2 and the Smc1 head (Scc2T1281C Smc1E1102C), between Scc2 and the Smc3 head (Scc2E819C Smc3S72C), and between Scc2 and Smc3's neck (Scc2D369C Smc3K1004C). As predicted by the cryo-EM structure, all three pairs enabled BMOE to crosslink Scc2 to SMC heads in the presence of ATP and DNA (*Figure 3A–C*). Crosslinking between Scc2 and either Smc1 or Smc3 head occurred in the absence of both ATP and DNA but was stimulated by ATP, an effect that was more pronounced for crosslinking between Scc2 and the Smc3 head. DNA also modestly increased crosslinking between both cysteine pairs, but only in the absence of ATP. In contrast, crosslinking between Scc2 and the Smc3 neck was strongly ATP dependent and enhanced by DNA (*Figure 3C*). These results suggest that Scc2 initially binds to the Smc1 head, subsequently binds the Smc3 head, and only binds the Smc3 neck efficiently upon the engagement of Smc1 and Smc3 heads in the presence of ATP and DNA.

We next created two different cysteine pair combinations to measure clamping using our DNA entrapment assay. The first combined Scc2D369C Smc3K1004C with Scc2E819C Smc3S72C, whose simultaneous crosslinking should entrap DNA in a covalent compartment formed by crosslinks between the N-terminal and central domains of Scc2 with Smc3's neck and head, respectively (Clamp; *Figure 3D*). The second combined Scc2D369C Smc3K1004C and Scc2T1281C Smc1E1102C with Smc1N1192C Smc3R1222C, a pair specific for engaged heads. Simultaneous crosslinking of all three interfaces should entrap DNA in a compartment created by Scc2's association with both Smc1 and

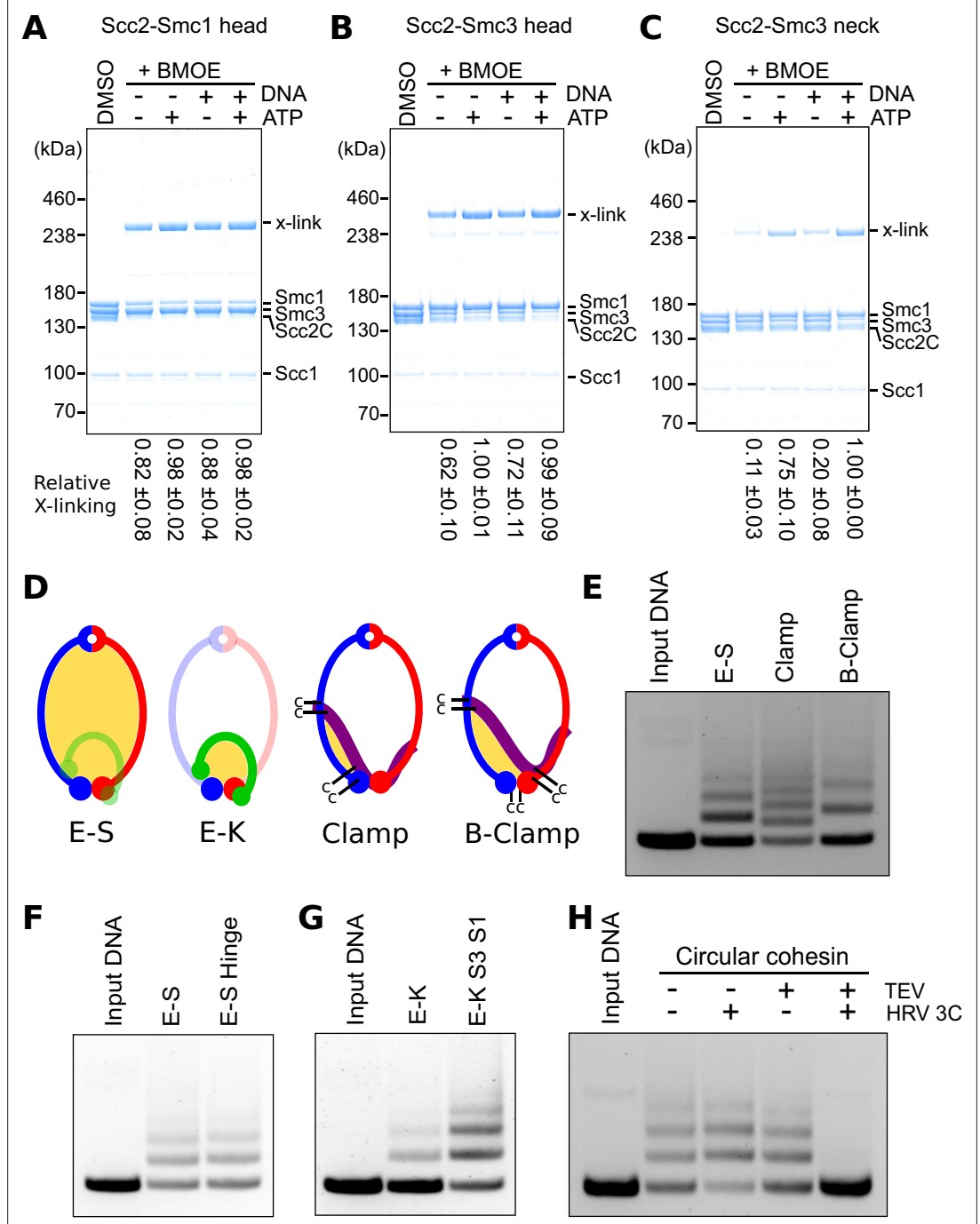

**Figure 3.** DNA passes through cohesin's ATPase domains. (**A**) Crosslinking Scc2 to the Smc1 head in the presence or absence of ATP and DNA. *n* = 3. (**B**) Crosslinking Scc2 to the Smc3 head in the presence or absence of ATP and DNA. *n* = 3. (**C**) Crosslinking Scc2 to the Smc3 neck in the presence or absence of ATP and DNA. *n* = 3. (**D**) Models of cohesin showing the E-S, E-K, Clamp, or below the clamp (B-Clamp) compartments, highlighted in yellow. For the Clamp and B-Clamp compartments Scc2 is in purple. (**E**) Entrapment of DNA in either the E-S, Clamp, or B-Clamp compartments in the presence of Scc2 after 2 min. (**F**) Entrapment of DNA in the E-S compartment by cohesin with either an open (E-S) or covalently closed (E-S Hinge) hinge in the presence of Scc2 after 2 min. (**G**) Entrapment of DNA in the E-K compartment by cohesin with either both kleisin interfaces open (E-K) or both covalently closed (E-K S3 S1) in the presence of Scc2 after 2 min. (**H**) Entrapment of DNA by covalently circular cohesin in the presence of Scc2 after 2 min. After crosslinking, BMOE was quenched by addition of dithiothreitol (DTT) and then the samples were treated with tobacco etch virus (TEV) and/or human rhinovirus (HRV) 3C proteases and incubated at 24°C for 30 min.

The online version of this article includes the following figure supplement(s) for figure 3:

**Figure supplement 1.** DNA passes through cohesin's ATPase domains.

**Figure supplement 2.** Entrapment in the E-S and E-K compartments occurs simultaneously.

Smc3 heads when they are engaged (below the Clamp or B-Clamp). Under the same conditions that promote entrapment in the E-S compartment, namely the presence of Scc2 and ATP, and the absence of Scc3, DNA was efficiently entrapped in both Clamp and B-Clamp compartments within 2 min (*Figure 3E*).

## Entrapping DNA in the E-S and E-K compartments involves passing DNA between ATPase head domains

DNA could enter the E-S compartment by passage 'down' through an opened hinge or 'up' between SMC heads (*Figure 3—figure supplement 1B*). To distinguish these, we analysed the effect of pre-sealing the hinge interface. This had no effect on E-S entrapment, excluding the possibility that DNA passes 'down' through the hinge (*Figure 3F*). It should be noted that passage through the hinge would result in S–K entrapment, which is not observed under these conditions (*Collier et al., 2020*). If DNA instead passes between the ATPase heads, without any dissociation of Scc1 from either the Smc3 neck or Smc1 head, then entrapment within the E-S compartment will be accompanied by entrapment between engaged heads and the kleisin subunit associated with them; that is in the E-K compartment, which we have previously shown (*Collier et al., 2020*). However, it could be argued that entrapment in the E-K compartment does not arise in this manner but rather as a result of a separate transport process in which DNA passes through a transiently opened SMC/kleisin interface either before or during head engagement. Such a mechanism has been invoked to explain clamping of DNA on top of engaged heads by Mis4, the *S. pombe* Scc2 ortholog (*Higashi et al., 2020*). To address whether kleisin disengagement is required for entrapment between engaged heads and their associated kleisin, we introduced the cysteine pair specific for engaged heads (Smc1N1192C Smc3R1222C) into the Smc3–Scc1–Smc1 fusion. Sealing both kleisin interfaces in this manner did not prevent entrapment within the E-K compartment. In fact, this construct entrapped DNA even more efficiently than WT (*Figure 3G*), presumably because only a single interface needs to be crosslinked by BMOE compared to the three required for WT. Clearly, entrapment within the E-K compartment in the presence of Scc2 does not involve passage through either Smc1/ or Smc3/kleisin gates. We conclude that the only interface that must open for entrapment of DNA within the E-S or E-K compartments is that between the Smc1 and Smc3 ATPase heads.

Due to their similar kinetics (*Collier et al., 2020*), it is likely that DNA entrapment in E-S and E-K compartments induced by Scc2 in the absence of Scc3 is a consequence of a single reaction, namely passage of DNA up between the head domains prior to their engagement without passage through any one of the ring's three interfaces. If true, cohesin with all three interfaces fused should still be able to entrap DNA in the E-S and E-K compartments. Furthermore, cleavage of either the SMC or kleisin moiety should have no effect on the amount of DNA entrapped, as DNA will remain entrapped within the remaining intact compartment (*Figure 3—figure supplement 2A*). Only simultaneous cleavage of both moieties should release DNA. To test this, we created a version of the covalently circular species of cohesin containing the cysteine pair necessary to crosslink the heads when engaged in the presence of ATP. A pair of tandem TEV protease cleavage sites were inserted in the linker connecting the spycatcher and the Smc3 hinge, which enables cleavage of the SMC moiety, while an HRV 3C protease site was present in the linker connecting Scc1 to Smc1, which enables cleavage of the kleisin moiety (*Figure 3—figure supplement 2B*). Incubating this construct with either TEV or HRV 3C leads to the linearization and opening of the E-S and E-K compartments, respectively, while incubation with both proteases leads to opening of both compartments, as well as release of a ~180 kDa digestion fragment comprised of the C-terminal half of Smc3 fused to Scc1 (*Figure 3—figure supplement 2C*). Circular cohesin was able to entrap DNA following the BMOE treatment that crosslinks engaged heads (*Figure 3H*) and remarkably entrapment was largely unaffected by incubation with either TEV or HRV 3C proteases after crosslinking. DNA was only released upon incubation with both proteases (*Figure 3H*). These results imply that DNA is simultaneously entrapped in both the E-S and E-K compartments due to a single transportation process that involves DNA passage between the Smc1 and Smc3 head domains prior to their engagement. Furthermore, these results allow us to infer the path of Scc1, which must pass over and 'above' the DNA, as has been suggested for condensin's Brn1 subunit (*Lee et al., 2022*).

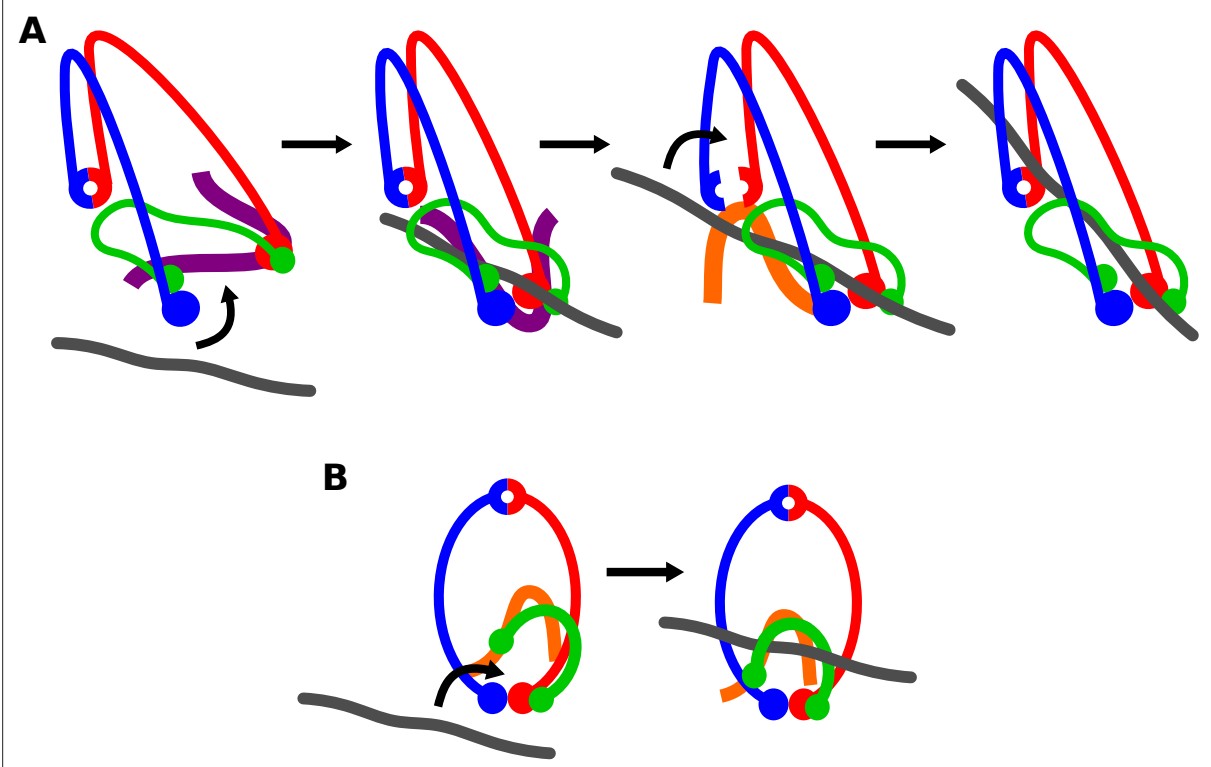

**Figure 4.** Models for DNA entry. (**A**) Model for DNA passage through the hinge. DNA passes 'up' through the ATPase heads and binds to Scc2 (purple). The SMC heads engage in the presence of ATP, and DNA is then clamped between Scc2 and the Smc3 neck. Scc3 (orange) is involved in opening and passing a downstream section of DNA through the hinge. (**B**) Model for DNA passage through the Smc3–Scc1 interface. ATP binding leads to the opening of the Smc3–Scc1 interface. DNA then binds to Scc3 and the interfaces closes.

## Discussion

In summary, we have shown that cohesin has two DNA gates, one at the hinge and a second at the Smc3/Scc1 interface. Available evidence suggests that the hinge gate is essential for the establishment of sister chromatid cohesion while the Smc3/Scc1 gate is not. Passage of DNA through the hinge is likely preceded by and very possibly dependent on its entrapment in a clamp between Scc2 and engaged ATPase heads (*Figure 4A*), a state created by passage of DNA between the SMC ATPase heads but not through either the hinge or Smc3/Scc1 gate. Following clamping, a section of DNA downstream of the clamp might then be passed through the hinge in a process dependent on Scc3. The precise role of Scc3 in hinge passage is currently unclear. It might either facilitate interface opening or the specific positioning of DNA against the hinge.

DNA entrapment through the Smc3/Scc1 interface presumably occurs when head engagement takes place in the absence of Scc2. Under these circumstances, kleisin N-terminal domains dissociate from SMC necks (*Muir et al., 2020*; *Lee et al., 2022*), creating either an entry or exit gate for DNA (*Figure 4B*). Our finding that entrapment via this interface does not require Scc2, unlike that through the hinge, is consistent with this scenario. Because entrapment by wild-type rings depends on Scc3's ability to bind DNA, we suppose that entrapment via the Smc3/Scc1 interface as well as the hinge involves this activity, though this remains to be tested directly. How binding of DNA to Scc3 facilitates passage remains unclear. It seems likely that passage of DNA through transiently opened Smc3/Scc1 interfaces will share properties with the process by which cohesin dissociates from chromosomes in vivo, which is also thought to depend on opening the Smc3/Scc1 interface (*Chan et al., 2012*; *Beckouët et al., 2016*). Whether entrapment through this interface takes place in vivo or if it instead reflects an in vitro artefact remains to be established.

Hitherto, our assay has only detected individual DNAs entrapped inside S–K rings. Entrapment of this nature occurs prior to DNA replication in vivo and is possibly converted to co-entrapment of sister DNAs during S phase with the help of specific replisome proteins (*Srinivasan et al., 2020*).

Crucially, conversion of cohesin that has associated with unreplicated DNA to a form that co-entraps sister DNAs does not require Scc2. If Scc2 is essential for passage of DNAs through the hinge, as our in vitro experiments suggest, then co-entrapment arising during conversion cannot involve any further passage of DNA through the hinge gate. It either involves the Smc3/Scc1 gate (neither Smc3/Scc1 opening nor conversion are essential) or arises from an activity that somehow pulls replicated DNAs through the ring without it being re-opened.

Our assay measuring entry through the hinge in vitro will enable the identification of mutants specifically defective in this process and these can subsequently be used to address whether passage of DNA through the hinge has roles in chromosome topology besides cohesion establishment, for example, in holding together TAD boundaries associated with convergent CTCF sites (*Liu and Dekker, 2021*). SMC hinge domains have two interfaces (north and south) and their dimerization creates a toroidal structure with a narrow lumen that is invariably positively charged (*Kurze et al., 2011*). Opening, either at one or both north and south interfaces (*Gruber et al., 2006*; *Shi et al., 2020*) would enable DNA to bind to highly conserved lysines residing inside the Smc1 hinge (*Srinivasan et al., 2018*), and this might be an important intermediate stage of the entrapment process. Whether the hinges of SMC complexes besides cohesin also act as DNA gates or whether their positively charged lumens merely bind DNA without passing it inside the ring is an important open question.

# Materials and methods

**Key resources table**

| Reagent type (species) or resource | Designation | Source or reference | Identifiers | Additional information |
|---|---|---|---|---|
| Recombinant DNA reagent | pACEbac1 Smc1 Smc3 Scc1-StrepII | This Study | pJC 93 | Plasmid obtainable from Nasmyth Lab |
| Recombinant DNA reagent | pACEbac1 Smc1 G22C K639C Smc3 E570C S1043C Scc1-StrepII A547C | This Study | pJC 95 | Plasmid obtainable from Nasmyth Lab |
| Recombinant DNA reagent | pACEbac1 Smc1 K639C N1192C Smc3 E570C R1222C Scc1-StrepII | This Study | pJC 153 | Plasmid obtainable from Nasmyth Lab |
| Recombinant DNA reagent | pACEbac1 Smc1 G22C N1192C Smc3 S1043C R1222C Scc1-StrepII A547C | This Study | pJC 154 | Plasmid obtainable from Nasmyth Lab |
| Recombinant DNA reagent | pACEbac1 Smc1 G22C K639C Smc3Scc1-StrepII E570C A547C | This Study | pJC 127 | Plasmid obtainable from Nasmyth Lab |
| Recombinant DNA reagent | pACEbac1 Smc1 G22C N1192C Smc3 S1043C R1222C Scc1-StrepII A547C | This Study | pJC 123 | Plasmid obtainable from Nasmyth Lab |
| Recombinant DNA reagent | pACEbac1 Smc1 G22C L597ST Smc3 S1043C S606SC Scc1-StrepII A547C | This Study | pJC 106 | Plasmid obtainable from Nasmyth Lab |
| Recombinant DNA reagent | pACEbac1 Smc3Scc1Smc1-StrepII E570C K639C | This Study | pJC 132 | Plasmid obtainable from Nasmyth Lab |
| Recombinant DNA reagent | pACEbac1 Smc1 G22C L597ST Smc3Scc1-StrepII S606SC A547C | This Study | pJC 129 | Plasmid obtainable from Nasmyth Lab |
| Recombinant DNA reagent | pACEbac1 Smc3 S1043C S606SC Scc1Smc1-StrepII L597ST | This Study | pJC 124 | Plasmid obtainable from Nasmyth Lab |
| Recombinant DNA reagent | pACEbac1 Smc3Scc1Smc1-StrepII S606SC TEV R1222C L597ST N1192C | This Study | pJC 135 | Plasmid obtainable from Nasmyth Lab |
| Recombinant DNA reagent | pACEbac1i Smc1-His E1102C Smc3 Scc1-StrepII | This Study | pJC 151 | Plasmid obtainable from Nasmyth Lab |
| Recombinant DNA reagent | pACEbac1i Smc1 Smc3 S72C Scc1-StrepII | This Study | pJC 100 | Plasmid obtainable from Nasmyth Lab |
| Recombinant DNA reagent | pACEbac1i Smc3 K1004C Smc1 _Scc1-StrepII | This Study | pJC 101 | Plasmid obtainable from Nasmyth Lab |
| Recombinant DNA reagent | pACEbac1i Smc1 Smc3 S72C K1004C Scc1-StrepII | This Study | pJC 102 | Plasmid obtainable from Nasmyth Lab |

*Continued on next page*

*Continued*

| Reagent type (species) or resource | Designation | Source or reference | Identifiers | Additional information |
|---|---|---|---|---|
| Recombinant DNA reagent | pACEbac1i Smc1 E1102C N1192C Smc3 K1004C R1222C Scc1-StrepII | This Study | pJC 105 | Plasmid obtainable from Nasmyth Lab |
| Recombinant DNA reagent | pACEbac1 Scc2$^{133-1493}$-StrepII | *Collier et al., 2020* | pJC 69 | Plasmid obtainable from Nasmyth Lab |
| Recombinant DNA reagent | pACEbac1 Scc2$^{133-1493}$-StrepII T1281C | This Study | pJC 75 | Plasmid obtainable from Nasmyth Lab |
| Recombinant DNA reagent | pACEbac1 Scc2$^{133-1493}$-StrepII E819C | This Study | pJC 77 | Plasmid obtainable from Nasmyth Lab |
| Recombinant DNA reagent | pACEbac1 Scc2$^{133-1493}$-StrepII D369C | This Study | pJC 76 | Plasmid obtainable from Nasmyth Lab |
| Recombinant DNA reagent | pACEbac1 Scc2$^{133-1493}$-StrepII D369C E819C | This Study | pJC 78 | Plasmid obtainable from Nasmyth Lab |
| Recombinant DNA reagent | pACEbac1 Scc2$^{133-1493}$-StrepII D369C T1281C | This Study | pJC 80 | Plasmid obtainable from Nasmyth Lab |
| Recombinant DNA reagent | pACEbac1 StrepII-Scc3 | *Collier et al., 2020* | pJC 84 | Plasmid obtainable from Nasmyth Lab |
| Chemical compound, drug | ATP | Sigma | Cat # 11140965001 | |
| Chemical compound, drug | BMOE | ThermoFisher | Cat # 22323 | |
| Chemical compound, drug | HRV 3 C protease | Pierce | Cat # 88946 | |
| Chemical compound, drug | TEV protease | Invitrogen | Cat # 12575023 | |
| Chemical compound, drug | EnzChek phosphate assay kit | Invitrogen | Cat # E6646 | |

## DNA and protein preparation

Protein and DNA components were prepared as described in *Collier et al., 2020*.

## ATPase assay

DNA was prepared in DNA buffer (25 mM 4-(2-hydroxyethyl)-1-piperazineethanesulfonic acid (HEPES) pH 7.5, 1 mM tris 2-carboxyethyl phopsphine (TCEP), and 5% glycerol) by annealing two complementary single-stranded 40 bp oligonucleotides by heating to 95°C for 5 min and decreasing in 0.1°C intervals every 15 s to a final temperature of 4°C. 150 µl reactions were prepared containing 50 nM cohesin (Smc1, Smc3, and Scc1), Scc3, Scc2, and 600 nM DNA in loading buffer (25 mM HEPES pH 7.5, 50 mM NaCl, 1 mM MgCl$_2$, 1 mM TCEP, and 5% glycerol), and EnzChek phosphate assay kit components (Invitrogen) added to their recommended concentrations. Reactions were initiated by the addition of ATP (Sigma) to a concentration of 1 mM. The ATPase reaction was followed by measuring the increase in absorbance at 360 nm over 60 min. Data shown an average of four experiments.

## DNA entrapment assay

13 µl reactions were prepared containing 165 nM cohesin (Smc1, Smc3, and Scc1) and 9.3 nM supercoiled pUC19 in loading buffer (25 mM HEPES pH 7.5, 50 mM NaCl, 1 mM MgCl$_2$, 1 mM TCEP, and 5% glycerol). When present, Scc3 was added to a concentration of 165 nM and Scc2 to a concentration of 55 nM. Reactions were initiated by the addition ATP (Sigma) to a concentration of 5 mM. Reactions were incubated at 24°C for either 40 or 2 min at 750 rpm. Crosslinking was carried out by addition of 1.5 µl BMOE (Thermo Scientific) to a concentration of 0.64 mM and incubated on ice for 6 min. Samples were then denatured by addition of 1.5 µl 10% SDS and then incubated at 70°C for 20 min at 750 rpm. DNA loading dye was then added and samples separated by agarose gel electrophoresis at 50 V for 17 hr at 4°C. Assays were repeated at least twice.

When cleaving the E-S and E-K compartments of circular cohesin 10 mM DTT was added after BMOE crosslinking and the samples incubated at 24°C for 5 min. Protein cleavage was carried out by addition of 1 µl TEV protease (Invitrogen) to cleave the E-S compartment and 1 µl HRV 3 C protease

(Pierce) to cleave the E-K compartment. To samples in which one or both proteases was omitted, loading buffer was added instead. Samples were then treated as in other experiments.

## Protein crosslinking assay

10 μl reactions were prepared containing 0.7 μM cohesin (Smc1, Smc3, and Scc1) and Scc2 in loading buffer (25 mM HEPES pH 7.5, 50 mM NaCl, 1 mM $MgCl_2$, 1 mM TCEP, and 5% glycerol). When present, ATP (Sigma) was added to a concentration of 10 mM and DNA (supercoiled pUC19) added to a concentration of 60 nM. Reactions were incubated at 24°C for 5 min and then either 1 μl DMSO added, or 1 μl BMOE (Thermo Scientific) added to a concentration of 0.64 mM. Samples were then denatured by addition of 4× lithium dodecyl sulfate (LDS) protein loading dye and heated at 70°C for 10 min. Samples were then separated by SDS–polyacrylamide gel electrophoresis (PAGE) using 3–8% Tris-acetate gels ran at 100 V for 4 hr 30 min at 4°C.

## Circular cohesin cleavage

10 μl samples were prepared containing 0.7 μM circular cohesin. Protein cleavage was carried out by addition of 1 μl TEV protease (Invitrogen) to cleave the E-S compartment and 1 μl HRV 3C protease (Pierce) to cleave the E-K compartment. To samples in which one or both proteases was omitted, loading buffer was added instead. Samples were then denatured by addition of 4× LDS protein loading dye and heated at 70°C for 10 min. Samples were then separated by SDS–PAGE using 3–8% Tris-acetate gels ran at 100 V for 4 hr 30 min at 4°C.

## Acknowledgements

We thank Mark Howarth (Oxford, UK) for his advice on the spycatcher system; Benedikt Bauer (IMP, Austria) for sharing information on the linkers to fuse the two SMC–kleisin interfaces; and Bin Hu (Aberdeen, UK) for sharing the cysteine pairs to crosslink Scc2 to both SMC heads. Finally, we thank all members of the Nasmyth and J Lowe groups for their invaluable discussions over the course of this work.

## Additional information

### Funding

| Funder | Grant reference number | Author |
| --- | --- | --- |
| Wellcome Trust | 107935/Z/15/Z | Kim A Nasmyth |
| Cancer Research UK | 26747 | Kim A Nasmyth |

The funders had no role in study design, data collection, and interpretation, or the decision to submit the work for publication. For the purpose of Open Access, the authors have applied a CC BY public copyright license to any Author Accepted Manuscript version arising from this submission.

### Author contributions

James E Collier, Conceptualization, Investigation, Methodology, Writing - original draft, Writing - review and editing; Kim A Nasmyth, Conceptualization, Funding acquisition, Investigation, Writing - original draft, Project administration, Writing - review and editing

### Author ORCIDs

James E Collier (i) http://orcid.org/0000-0002-9904-9423
Kim A Nasmyth (i) http://orcid.org/0000-0001-7030-4403

### Decision letter and Author response

Decision letter https://doi.org/10.7554/eLife.80310.sa1
Author response https://doi.org/10.7554/eLife.80310.sa2

## Additional files

### Supplementary files
• MDAR checklist

• Source data 1. All source data from the study.

### Data availability
All data generated or analysed during this study are included in the Source Data file.

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
