## [Editor Report]

This paper is a fundamental step forward in our understanding of how cohesin interacts with DNA to execute sister chromatid cohesion. Compelling biochemical evidence is presented to demonstrate the functional entry gate of cohesin for the topological entrapment of DNA. This important study will be of interest to researchers in the chromosome biology field, particularly those focused on SMC proteins and genome organisation.

---

## [Decision Letter]

**Decision letter after peer review:**

Thank you for submitting your article "DNA passes through cohesin's hinge as well as its Smc3-kleisin interface" for consideration by *eLife*. We apologise for the delay in the review process.

Your article has been reviewed by 2 peer reviewers, and the evaluation has been overseen by a Reviewing Editor and Jessica Tyler as the Senior Editor. The following individual involved in the review of your submission has agreed to reveal their identity: Hongtao Yu (Reviewer #2).

Essential revisions:

The data presented is convincing and important. Please address the requested revisions from the reviewers below to improve the clarity of the manuscript for the non-expert. The most important are:

1. A better description of the assay in the text. Please also include the schematic describing the assay in the main figures. Please also label the different species on the gels assaying entrapment (Figure 2 etc.).

2. Please clearly define, ideally with the inclusion of schematics, the distinctions between "SMC", "kleisin", and "S-K" compartments.

3. Figure 3A-C. Please label the figure to indicate which interfaces are being assayed.

*Reviewer #1 (Recommendations for the authors):*

1. Biochemical data: Figure 2A: The authors need to clarify that the different fusion proteins generated are made in the context of the 6C version.

2. Line 101 et seq: Restate that the entrapment assay used involves BMOE crosslinking and SDS denaturation followed by agarose gel electrophoresis. Clarify that the assay enables the detection of covalently closed cohesin rings.

3. Not all readers will be familiar with Collier et al., 2020. To facilitate the understanding of how these experiments are done, I would recommend moving panel A in Figure supplement 2 to the main Figure 2A (unless Figure supplement 2 is presented alongside Figure 2 in the online version).

4. Figure 2E: It is not fully clear why in the hinge fusion DNA cannot pass through the Smc1/Scc1 interface as BMOE treatment is done following entrapment. If so, one cannot conclude that the entrapment seen occurs uniquely through the Smc3/Scc1 interface?

5. Figure Supplement 3F: Can the authors indicate in the schematic the 'kleisin' and 'SMC' compartment (maybe using the same color code as in Figure Supplement 3A)? It would be helpful to clarify why the opening of both compartments is required to abolish entrapment. Presumably, if the kleisin passes over and above the DNA, entrapment would simply depend on SMC3/SMC1 head interface engagement, at least in the absence of covalent closure/crosslinking?

6. Please clarify the model in Figure 4: If entrapment through the Smc3/Scc1 interface requires Scc3 but not Scc2 (Figure 2E), it is unclear why the model proposes that Scc3 is involved in the opening and transport of DNA through the hinge. Clarify why the Discussion section states that: 'Scc2 is essential for passage of DNAs through the hinge…'.

7. The paper would benefit from condensation.

---

## [Author Response]

Essential revisions:The data presented is convincing and important. Please address the requested revisions from the reviewers below to improve the clarity of the manuscript for the non-expert. The most important are:1. A better description of the assay in the text. Please also include the schematic describing the assay in the main figures. Please also label the different species on the gels assaying entrapment (Figure 2 etc.).

Done. Figure 2a.

2. Please clearly define, ideally with the inclusion of schematics, the distinctions between "SMC", "kleisin", and "S-K" compartments.

Done. Figure 2b. Figure 3d

3. Figure 3A-C. Please label the figure to indicate which interfaces are being assayed.

Done.

Reviewer #1 (Recommendations for the authors):1. Biochemical data: Figure 2A: The authors need to clarify that the different fusion proteins generated are made in the context of the 6C version.

Done.

2. Line 101 et seq: Restate that the entrapment assay used involves BMOE crosslinking and SDS denaturation followed by agarose gel electrophoresis. Clarify that the assay enables the detection of covalently closed cohesin rings.

Done.

3. Not all readers will be familiar with Collier et al., 2020. To facilitate the understanding of how these experiments are done, I would recommend moving panel A in Figure supplement 2 to the main Figure 2A (unless Figure supplement 2 is presented alongside Figure 2 in the online version).

Done.

4. Figure 2E: It is not fully clear why in the hinge fusion DNA cannot pass through the Smc1/Scc1 interface as BMOE treatment is done following entrapment. If so, one cannot conclude that the entrapment seen occurs uniquely through the Smc3/Scc1 interface?

While it is still technically possible that by only fusing the hinge DNA could be passing through either kleisin interface, we demonstrate in figure 2D that cohesin with both the hinge and Smc3-Scc1 interface closed does not entrap DNA. From this we conclude that DNA does not pass through the Scc1-Smc1 interface.

5. Figure Supplement 3F: Can the authors indicate in the schematic the 'kleisin' and 'SMC' compartment (maybe using the same color code as in Figure Supplement 3A)? It would be helpful to clarify why the opening of both compartments is required to abolish entrapment. Presumably, if the kleisin passes over and above the DNA, entrapment would simply depend on SMC3/SMC1 head interface engagement, at least in the absence of covalent closure/crosslinking?

Added cartoon Figure 3 —figure supplement 2a to help clarify this.

6. Please clarify the model in Figure 4: If entrapment through the Smc3/Scc1 interface requires Scc3 but not Scc2 (Figure 2E), it is unclear why the model proposes that Scc3 is involved in the opening and transport of DNA through the hinge. Clarify why the Discussion section states that: 'Scc2 is essential for passage of DNAs through the hinge…'.

Scc2 is required for hinge passage as shown in Figure 2E. We believe passage through the hinge is an activity directly mediated by Scc3, which follows Scc2 driven clamp formation. Scc2’s role in hinge passage may just be to form this prior clamped state. We propose Scc3 also has roles in passing DNA through the Smc3/Scc1 interface and this activity is distinct from that of passage through the hinge.